# AutoPrune: Automatic Network Pruning by Regularizing Auxiliary Parameters

**Xia Xiao, Zigeng Wang, Sanguthevar Rajasekaran**[*]
Department of Computer Science and Engineering
University of Connecticut
Storrs, CT, USA, 06269
{xia.xiao, zigeng.wang, sanguthevar.rajasekaran}@uconn.edu

## Abstract

Reducing the model redundancy is an important task to deploy complex deep learning models to resource-limited or time-sensitive devices. Directly regularizing or modifying weight values makes pruning procedure less robust and sensitive to the choice of hyperparameters, and it also requires prior knowledge to tune different hyperparameters for different models. To build a better generalized and easy-to-use pruning method, we propose AutoPrune, which prunes the network through optimizing a set of trainable auxiliary parameters instead of original weights. The instability and noise during training on auxiliary parameters will not directly affect weight values, which makes pruning process more robust to noise and less sensitive to hyperparameters. Moreover, we design gradient update rules for auxiliary parameters to keep them consistent with pruning tasks. Our method can automatically eliminate network redundancy with recoverability, relieving the complicated prior knowledge required to design thresholding functions, and reducing the time for trial and error. We evaluate our method with LeNet and VGG-like on MNIST and CIFAR-10 datasets, and with AlexNet, ResNet and MobileNet on ImageNet to establish the scalability of our work. Results show that our model achieves state-of-the-art sparsity, e.g. 7%, 23% FLOPs and 310x, 75x compression ratio for LeNet5 and VGG-like structure without accuracy drop, and 200M and 100M FLOPs for MobileNet V2 with accuracy 73.32% and 66.83% respectively.

## 1 Introduction

Deep neural networks (DNNs) have achieved a significant success in many applications, ranging from image classification He *et al.* [2016] and object detection Ren *et al.* [2015] to self driving Maqueda *et al.* [2018] and machine translation Sutskever *et al.* [2014]. However, the computationally expensive and memory intensive properties of DNNs prevent their direct deployment to devices such as mobile phones and auto-driving cars. To overcome these challenges, learning compressed light-weight DNNs has attracted growing research attention Han *et al.* [2015]; Dong *et al.* [2017]; Zhuang *et al.* [2018].

For recent pruning methods, prior knowledge plays an important role in improving the performance and reducing the training time, in which a large number of hyperparameters need to be individually designed for different architectures and datasets. In magnitude-based pruning, where weights lower than thresholds will be removed, the chosen thresholds majorly affect the pruning performance Han *et al.* [2015]; Guo *et al.* [2016]. Moreover, for the layer-wise pruning Dong *et al.* [2017]; Aghasi *et al.* [2017], the searching space for layer-wise threshold combinations can be exponential in the number of layers. As another branch of pruning, sensitivity-based method Tartaglione *et al.* [2018]

---

[*]Corresponding author. This work has been supported in part by the following NSF grants: 1447711, 1514357, 1743418, and 1843025.

removes the less sensitive weights from the network, while further hyperparameter/function design is required to avoid undesired weight shrinkage or updates.

Recently research on pruning Liu *et al.* [2019b] implies that the pruning process is actually finding the right network structure, thus bridging the gap between pruning and neuron architecture search(NAS). However, state-of-art NAS methods cannot be directly applied to pruning task. For example, gradient based search algorithm DART Liu *et al.* [2019a] introduces auxiliary parameters acting as indicators to select the appropriate network structure optimized through a gradient-descent procedure. But, discrepancy between continuous over-parameterized graph and the discretized sub-graph is unavoidable during the model evaluation procedure, and zero operation is eliminated in the search space. Our method is similar to DART such that we employ smooth, approximated, gradient-based search to pruning task, but the discrepancy is reduced by iteratively evaluating recoverable sub-graph during the pruning procedure.

The advantage of introducing auxiliary parameters to pruning task is hyperparameter insensitive. Instead of directly regularizing weights, our method regularizes auxiliary parameters which aggregate gradient perturbations such as batch noise, dead neuron or dropout during pruning. In this way, temporarily incorrect pruning induced by the instability and non-optimal hyperparameters can be recovered, which greatly contributes to the pruning performance and efficiency. Different from updating auxiliary parameters with vanilla unstable linear coarse gradient in Srinivas *et al.* [2017], in order to stabilize the pruning procedure, we analyze and decouple the gradient between weight parameters and auxiliary parameters. In contrast to Louizos *et al.* [2018], our method avoids inefficient and high variance single-step Monte-Carlo sampling and places no assumptions on the prior distribution. In comparison with Carreira-Perpinán and Idelbayev [2018], we add no constraints on model parameters, maintaining the flexibility and capacity of the model. In addition, we design a sparse regularizer working with the original loss function and weight decay. In order to evaluate the proposed method, we conduct extensive experiments on different datasets and models, and the results show that our method achieves state-of-the-art performance.

Contributions and novelty of our work are: 1) we offer a gradient based automatic network pruning model; 2) we propose novel and weakly coupled update rule for auxiliary parameters to stabilize pruning procedure; 3) we reduce the sub-graph discrepancy by iteratively evaluating recoverable sub-graph; 4) we evaluate different smooth approximations of the derivative of the rectifier; 5) we obtain the state-of-art results on both structure and weight pruning and our method is scalable on modern models and datasets.

## 2   Related Work

Neural network pruning can be mainly classified into two categories: unstructured pruning and structured pruning. Unstructured pruning compresses neural networks by dropping redundant/less-meaningful weights, while structured pruning is by dropping neurons. Both pruning methods shrink the storage space of the targeted neural network, but, comparatively speaking, structured pruning has a directly benefit in reducing the computational cost of DNNs.

LeCun *et al.* [1990] pioneers neural network pruning and proposes optimal brain damage method for shallow neural network unstructured pruning. For DNNs, Han *et al.* [2015] presents global magnitude-based weight pruning and Guo *et al.* [2016] introduces recoverability into the global pruning. Similar idea has then been applied to structured pruning. Hu *et al.* [2016] removes neurons with high average zero output ratio and Li *et al.* [2017] prunes neurons with low absolute summation values of incoming weights, which are all replying on predefined thresholds.

In order to further improve the compression rate, different layer-wise pruning methods have been proposed, either by weighting connections based on a layer-wise loss function(Dong *et al.* [2017]) or by solving a specially designed convex optimization program(Aghasi *et al.* [2017]). These layer-wise schemes provide theoretical error bounds for specific activation functions but leave many hyperparameters to be carefully designed. Due to this issue, Li *et al.* [2018] presents a relatively efficient comprehensive optimization algorithm for tuning layer-wise hyperparameters.

Besides layer-wise schemes, Gordon *et al.* [2018] scales efficient structured pruning on large networks by applying resource weighted sparsifying regularizers on activations. Zhu *et al.* [2018] improves neural network sparsity by explicitly forcing the network to learn a set of less correlated filters via

decorrelation regularization. Zhuang *et al.* [2018] designs a discrimination-aware channel pruning method to locate most discriminative channels. But after ranking the filters or channels, we still have to pinpoint their optimal combinations for each layer, which highly relies on expertise. Gomez *et al.* [2019] proposed to keep neurons with high magnitude and prune neurons with smaller magnitude in a stochastic way. The accuracy is maintained by reducing the dependency of important neurons on unimportant neurons.

Liu *et al.* [2019b] does comprehensive experiments showing that training-from-scratch on the right sparse architecture yields better results than pruning from pre-trained models. Searching for a spare architecture is more important than the weight values. Liu *et al.* [2019a] employs continuous indicator parameters to relax the non-differentiable architecture searching problem. The relaxation is then removed by dropping weak connections and selecting the single choice of the k options with the highest weight. However, the gap between the continuous solution and the discretized architecture remain unknown. More importantly, zero operations are omitted during the derivation process, making is unsuitable for network pruning. Yu and Huang [2019a] implements greedy search for width multipliers of slimmable network(Yu *et al.* [2018]) to reduce kernel number. Multiple batch normalization layers are trained under different channel settings. However, a significant accuracy drop is observed in extreme sparse cases.

## 3 Methods

In this section, we first formulate the problem and discuss the indicator function and auxiliary parameters. Then, we introduce the update rule for auxiliary parameters for stable and efficient network pruning. Without losing generality, our method is formulated on weight pruning, but it can be directly extended to neuron pruning.

### 3.1 Problem Formulation

Let $f_w : \mathcal{R}^{m \times n} \to \mathcal{R}^d$ be a continuous and differentiable neural network parametrized by $\mathcal{W}$ mapping input $X \in \mathcal{R}^{m \times n}$ to target $Y \in \mathcal{R}^d$. The pruning problem can be formulated as:

$$\underset{w}{\mathrm{argmin}} \; \frac{1}{N} \left( \sum_{i=1}^{N} \mathcal{L}(f(x_i, W), y_i) \right) + \mu ||W||_0, \tag{1}$$

where $||W||_0$ denotes zero norm, or number of non-zero weights. The goal is to find the sparse architecture with minimum subset $w \in \mathcal{W}$ that preserves the model accuracy. However, the second term is non-differentiable, making the problem not solvable using gradient descent. Direct regularization on $w_{ij}$ will lead to sensitivity on hyperparameter $\mu$ and instability with batched training. We relax this problem by introducing a indicator function defined as:

$$h_{ij} = \begin{cases} 0, & \text{if } w_{ij} \text{ is pruned;} \\ 1, & \text{otherwise.} \end{cases} \tag{2}$$

Instead of designing an indicator function for each $w_{ij}$ manually, we propose to parameterized a universal indicator function by a set of trainable auxiliary parameters $M$. Due to the non-differentiable property of the indicator function, we will discuss how to update auxiliary parameters in subsections 3.2 and 3.3. Then the network sparsification problem can be re-formulated as an optimization problem:

$$\underset{w,m}{\mathrm{argmin}} \; \frac{1}{N} \left( \sum_{i=1}^{N} \mathcal{L}(f(x_i, W \odot h(M)), y_i) \right) + \lambda \mathcal{R}(W) + \mu \mathcal{R}(h(M)), \tag{3}$$

where $\mathcal{R}(\cdot)$ denotes a regularization function. We also denote the element-wise product $T = W \odot h(M)$ as the weight matrix after pruning. The advantage of regularizing on auxiliary parameters instead of original weights is that any change in $m_{ij}$ does not directly influence the gradient update of $w_{ij}$, leading to a less sensitive pruning process with respect to hyperparameter $\mu$.

As done by Han *et al.* [2015] and Carreira-Perpinán and Idelbayev [2018], in order to enhance the stability and performance, we also implement a multi-step training through iteratively training the sparsity structure and retraining the original weights. More specifically, we employ the bi-level optimization used in Liu *et al.* [2019a] for the optimization problem. The training set will be split

into $X_{train}$ and $X_{val}$, and we can further re-formulate the problem from minimizing a single loss function to minimizing the following loss functions iteratively.

$$\min_w \mathcal{L}_1 = \min_w \sum_{i=1}^{N} \mathcal{L}(f(x_i, W \odot h(M)), y_i) + \lambda \mathcal{R}(W), \ x_i \in X_{train}, \tag{4}$$

$$\min_m \mathcal{L}_2 = \min_m \sum_{i=1}^{N} \mathcal{L}(f(x_i, W \odot h(M)), y_i) + \mu \mathcal{R}(h(M)), \ x_i \in X_{val}, \tag{5}$$

The first term in both loss functions is the regular accuracy loss for neural network training. Note that the regularization of $W$ is not necessarily required but we add the term to show that our method is consistent with traditional regularizers.

## 3.2 Coarse Gradient for Indicator Function

The indicator function $h_{ij}$ contains only zero and one values and thus is non-smooth and non-differentiable. Inspired by Hubara *et al.* [2016] where binary weights are represented using step functions and trained with hard sigmoid straight through estimator (STE), we use a simple step function for indicator function $h_{ij}$ with trainable parameter $m_{ij}$.

Binarized neural networks (BNNs) with proper STE have been demonstrated to be quite effective in finding optimal binary parameters and can achieve promising results in complex tasks. The vanilla BNNs are optimized by updating continuous variables $m_{ij}$:

$$\frac{\partial \mathcal{L}}{\partial m_{ij}} = \frac{\partial \mathcal{L}}{\partial \sigma(m_{ij})} \frac{\partial \sigma(m_{ij})}{\partial m_{ij}}, \ where \ \ \sigma(m_{ij}) = max(0, min(1, \frac{m_{ij}+1}{2})). \tag{6}$$

The output of each weight is the output of the hard sigmoid binary function. Note that the gradients of $\frac{\partial \sigma(m_{ij})}{\partial m_{ij}}$ can be estimated in multiple ways.

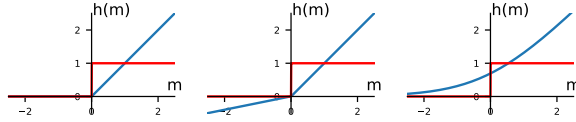

Figure 1: Coarse Gradients for STEs

Srinivas *et al.* [2017] discuss using BNNs to learn sparse networks, however, the authors suggest using linear STE to quickly estimate the gradient of the heaviside function. Recent result Yin *et al.* [2019] shows that ReLU or clipped ReLU STEs yield better convergence while linear STE is unstable at minima. Unfortunately, as shown in Fig. 1, the gradient of ReLU is zero if the input $m$ is smaller than zero. In other words, if we apply auxiliary parameters directly to any weight, without any regularization, the weight will permanently *die* once the corresponding weight has been pruned. Considering the pruning recoverability, we suggest using Leaky ReLU or Softplus instead of ReLU.

## 3.3 Updating Auxiliary Parameters

Instead of directly applying the gradient update as described in Eq. 6, we propose a modified update rule of auxiliary parameters to be consistent with (1) the magnitude of weights; (2) the change of weights; and (3) the directions of BNN gradients. The update rule of $m_{ij}$ is defined as:

$$m_{ij} := m_{ij} - \eta \left( \frac{\partial \mathcal{L}_{acc}}{\partial t_{ij}} sgn(w_{ij}) \frac{\partial h(m_{ij})}{\partial m_{ij}} \right) - \mu \frac{\partial h(m_{ij})}{\partial m_{ij}} \tag{7}$$

where $\mathcal{L}_{acc}$ denotes $\mathcal{L}(f(x_i, W \odot h(M)), y_i)$, $\eta$ is the learning rate of $m_{ij}$, $t_{ij} = w_{ij} \odot h(m_{ij})$, the second term can be considered as the gradient of $m_{ij}$, $\frac{\partial t_{ij}}{\partial m_{ij}}$, and the third term is related to the sparse regularizer. The proposed update rule is motivated from three advantages:

**Sensitivity Consistency:** The gradient of a vanilla BNN is correlated with $w_{ij}$, i.e., $\frac{\partial \mathcal{L}_{acc}}{\partial m_{ij}} \propto \frac{1}{f(|w_{ij}|)}$, which means that $m_{ij}$ is more sensitive if the magnitude of the corresponding $w_{ij}$ is large. Such a sensitive correlation is counter-intuitive since a larger $w_{ij}$ is more likely to be pruned with a small turbulence which reduces the robustness of the pruning. In the proposed update rule, we decouple

such a correlation to increase the stability of the pruning procedure. Practically, in order to boost the sensitivity of $m_{ij}$ associated with smaller weight magnitude(i.e. sensitivity consistency), we use a multiplier $w_{ij}$ to Eq. 7.

**Correlation Consistency:** The second advantage of the update rule is that the direction of the gradient of an arbitrary auxiliary parameter $m_{ij}$ is the same as the direction of the gradient of its corresponding $|w_{ij}|$, when ignoring the regularizers, i.e., $sgn(\frac{\partial \mathcal{L}_2}{\partial m_{ij}}) = sgn(\frac{\partial \mathcal{L}_1}{\partial |w_{ij}|})$.

**Proof.** We can expand the gradient for $w_{ij}$ and $m_{ij}$ as follows:

$$\frac{\partial \mathcal{L}_1}{\partial w_{ij}} = \frac{\partial \mathcal{L}_{acc}}{\partial t_{ij}} \frac{\partial t_{ij}}{\partial w_{ij}} + \lambda \frac{\partial \mathcal{R}(w_{ij})}{\partial w_{ij}} = \frac{\partial \mathcal{L}_{acc}}{\partial t_{ij}} h(m_{ij}) + \lambda \frac{\partial \mathcal{R}(w_{ij})}{\partial w_{ij}} \tag{8}$$

$$\frac{\partial \mathcal{L}_2}{\partial m_{ij}} = \frac{\partial \mathcal{L}_{acc}}{\partial t_{ij}} \frac{\partial t_{ij}}{\partial m_{ij}} + \mu \frac{\partial \mathcal{R}(h(m_{ij}))}{\partial m_{ij}} = \frac{\partial \mathcal{L}_{acc}}{\partial t_{ij}} w_{ij} \frac{\partial h(m_{ij})}{\partial m_{ij}} + \mu \frac{\partial h(m_{ij})}{\partial m_{ij}} \tag{9}$$

If we consider the direction of the first term of both gradients while ignoring the regularizers:

$$sgn(\frac{\partial \mathcal{L}_1}{\partial w_{ij}}) = sgn(\frac{\partial \mathcal{L}_{acc}}{\partial t_{ij}}) sgn(h(m_{ij}))$$
$$sgn(\frac{\partial \mathcal{L}_2}{\partial m_{ij}}) = sgn(\frac{\partial \mathcal{L}_{acc}}{\partial t_{ij}}) sgn(w_{ij}) sgn(\frac{\partial h(m_{ij})}{\partial m_{ij}}). \tag{10}$$

Given the conditions that $h(m_{ij}) \geq 0$ and $\frac{\partial h(m_{ij})}{\partial m_{ij}} \geq 0$, we can conclude that

$$sgn(\frac{\partial \mathcal{L}_2}{\partial m_{ij}}) = sgn(\frac{\partial \mathcal{L}_1}{\partial |w_{ij}|}). \tag{11}$$

∎

In other words, the auxiliary parameter $m_{ij}$ tracks the changing of the magnitude of $w_{ij}$. For the pruning task, when the absolute value of a weight/neuron keeps moving towards zero, we should accelerate the pruning process of the weight/neuron.

**Direction Consistency:** The third advantage of the update rule is that the inner product between the expected coarse and population gradients with respect to $m$ is greater than zero, i.e., the update gradient and the population gradient form an acute angle. Updating in this way actually reduces the loss of vanilla BNNs. We refer to Eq. 5, Lemma4 and Lemma10 from Yin *et al.* [2019], where the ReLU and linear STE form acute angle with population gradient. Since $\langle g_\sigma, g \rangle = \sigma' q(w, w*)$, where $q(w, w*)$ is a deterministic function for both cases and $\sigma$ represent the STE function. Since $\sigma'_{relu} \leq \sigma'_{LeakyRelu} \leq \sigma'_{Linear}$, we can then retain $0 \leq \langle g_{relu}, g \rangle \leq \langle g_{LeakyRelu}, g \rangle \leq \langle g_{Linear}, g \rangle$.

### 3.4 Recoverable Pruning

Pruning with recoverability is important to reduce the gap between the original network graph and the sparse network graph, which helps to achieve better sparsity. We design the pruning step following the idea of Dynamic Network Surgery(Guo *et al.* [2016]), that once some important weights are pruned and a large discrepancy occurs, the *incorrectly* pruned weights will be recovered to compensate for the increase of loss. Different from previous works with hard thresholding, for a specific weight/neuron, its opportunity to be pruned is determined automatically during optimization. The pruning step in our model is soft, the pruned weight will hold its value, and ready to be spliced back to the network if large discrepancy is observed.

Based on the multi-step training framework, after $m_{ij}$ is updated by Eq. 7, the unpruned network parameters $w_{ij}$ will be updated based on the newly learned structure. If no regularization is applied on $w_{ij}$, the corresponding $m_{ij}$ could be recovered by the accuracy loss. Note that a weight will be recovered if the damage made by the pruned weight cannot be restored by updating other unpruned weights. If weight decay is applied, any pruned weight will gradually lose recoverability with a fixed rate. The weight decay will decrease the magnitude of $w_{ij}$ and provide a negative gradient to $m_{ij}$, which reduces the recoverability. Whether a weight will be recovered under weight decay depends on (1) the absolute value of $w_{ij}$, and (2) the damage it made when removing it from the network. More specifically, recovering a weight $w_{ij}$ requires the gradient of $m_{ij}$ moving toward positive direction. With L1 regularization, a weight will be permanently pruned when its absolute value drops to zero.

---

**Algorithm 1** AutoPrun

| | |
|---|---|
| **Input**: Data set $X$ and $iter$ | 7: **while** iter!=0 **do** |
| **Parameter**: $W$, $M$, $\lambda$ and $\mu$ | 8:    Sample a mini batch from $X_{val}$; |
| **Output**: Auxiliary parameter $M$ and $W$ | 9:    Compute $grad_w$ with $L_1$; |
| | 10:   Compute $grad_m$ and $grad_{mr}$ by Eq. 7; |
| 1: Randomly split $X$ into $X_{train}$ and $X_{val}$. | 11:   Update $M$ with $grad_m$ and $grad_{mr}$; |
| 2: **if** Pre-trained **then** | 12:   Sample a mini batch from $X_{train}$; |
| 3:    Initialize $M$ based on pre-trained $W$; | 13:   Compute $grad_w$ with $L_1$; |
| 4: **else** | 14:   Update $W$ with $grad_w$; |
| 5:    Initialize $M \sim Gaussian(\mu, \sigma^2)$; | 15:   Update iter, $\lambda$, $\mu$ (if scheduling); |
| 6: **end if** | 16: **end while** |
| | 17: **return** solution; |

---

### 3.5 Acceleration by Regularizers

#### 3.5.1 Sparse Regularizer

Without any regularizer, our model can gradually converge to a sparse model, but with relatively slow speed, especially when the weights are close to optimal and the gradients with respect to $T = W \odot h(M)$ are almost zero. In order to accelerate the pruning process, we bring in regularizers to force the mask values to approach zero. The sparse regularizer is defined as:

$$\mathcal{R}(h(M)) = \sum_{i,j} |h(m_{ij})| = count(h(M)). \tag{12}$$

Note that the $L1$ regularizer applied on $h(M)$ directly counts the number of gates that are open, which is equivalent to applying L0 regularizer on $h(M)$. With the regularizer, $M$ will be pushed towards zero since the gradient with respect to $m_{ij}$ is the positive STE gradient. Another benefit of this regularizer is to filter out the noise when updating $W$ with SGD or dropout, i.e., $\mu \partial L_2 / \partial m_{ij} > 0$ when $\Delta |w| < \delta$ and $m_{ij}$ still decreases when $w_{ij}$ increases by only a small amount.

#### 3.5.2 Working with Weight Decay Regularizer

Our model can also work with general 1-norm or 2-norm regularizers on weights $W$. Since the auxiliary parameters $M$ follow $|W|$, any weight decay regularizer will help to increase the sparsification speed. An important side effect of weight decay regularizer is that after pruning a certain weight, the only source that can change $|w_{ij}| \in |W|$ $s.t.$ $h(m_{ij}) = 0$ will be the weight decay regularizers. A large weight decay hyperparameter will decrease the pruned weight fast and hamper the recoverability discussed in the previous subsection.

### 3.6 Hyperparameters Sensitivity and Robustness

By proposing auxiliary parameters and an indicator function, we introduce two new hyperparameters, learning rate hyperparameter $\eta$ and regularization hyperparameter $\mu$. However, the pruning procedure is not sensitive to those hyperparameters based on the following reasons: 1) We are not directly regularizing $W$, so the bias of STE and hyperparameter will not directly influence weights; 2) The indicator function is tolerant to the turbulence of auxiliary parameters $m_{ij}$; and 3) The pruning is recoverable when an incorrect pruning happens and the damage is made. Practically, as shown in the experimental part, the learning rate $\eta$ is scheduled to be the same as for learning the original weights $W$, and the regularization hyperparameter $\mu$ is set to be the same in all test cases. To conclude, our method reduces a set of hyperparameters to one single, non-sensitive hyperparameter.

### 3.7 Convergence Discussion

Similar to Gordon *et al.* [2018], our framework doesn't guarantee convergence when optimized with regularizers. But since the sparsification procedure is emperically fast and a good structure can be obtained with fewer epochs, we do not always need to wait until convergence. But, in order to give a guidance to hyperparameter tuning, we will briefly discuss the necessary condition for convergence. At convergence, if no regularization is applied, $\frac{\partial \mathcal{L}_{acc}}{\partial t_{ij}} = 0$. We can further conclude:

$$\frac{\partial \mathcal{L}_{acc}}{\partial t_{ij}} h(m_{ij}) = \frac{\partial \mathcal{L}_{acc}}{\partial t_{ij}} sgn(w_{ij}) \frac{\partial h(m_{ij})}{\partial m_{ij}} = 0. \tag{13}$$

If both weight decay and sparse regularizers are applied, we need $\frac{\partial \mathcal{L}_1}{\partial w_{ij}} = \frac{\partial \mathcal{L}_2}{\partial m_{ij}} = 0$. Assuming that pruned weights are sufficiently small and make no contribution to both gradients, we only consider the gradients w.r.t. $m_{ij} \in M$ $s.t.$ $m_{ij} > 0$, and $h(M) = 1$. When taking into account the learning rate compensation, we have:

$$0 = \frac{\partial \mathcal{L}_{acc}}{\partial t_{ij}} + \lambda \frac{\partial \mathcal{R}(w_{ij})}{\partial w_{ij}} = \frac{\partial \mathcal{L}_{acc}}{\partial t_{ij}} sgn(w_{ij}) \frac{\partial h(m_{ij})}{\partial m_{ij}} + \mu \frac{\partial h(m_{ij})}{\partial m_{ij}}. \qquad (14)$$

If L2 is applied, we have the necessary condition $2\lambda|w_{ij}| = c\mu$, where $c$ is the non-linear factor by different STEs. If L1 is applied, we have the necessary condition $\lambda = c\mu$. Under both cases, $\lambda$ and $\mu$ should be reduced to the same level when convergence.

## 4 Experiments

In this section, we introduce our experiment settings, and compare the neuron pruning and weight pruning performance with existing approaches.

### 4.1 Settings

To ensure a fair comparison, we follow the same backend packages as described in other papers. Except for LeNets, all the other pre-trained parameters are downloaded from commonly available sources and the auxiliary parameters are either initialized randomly or by pre-trained weights. All the accuracy results are the average of 10 runs and the spare structure is picked from the best accurate model. Our models are implemented by Tensorflow and run on Ubuntu Linux 16.04 with 32G memory and a single NVIDIA Titan Xp GPU. To show the insensitivity of the introduced hyperparameter, we set the learning rate of auxiliary parameters to 1.5e-2 and $\mu$ to 5e-2 for all test cases.

Table 1: Comparison of Different Neuron Pruning Techniques

| Model | Methods | Base Error | Error | Epochs | Neurons per Layer | NCR | FLOPs |
|---|---|---|---|---|---|---|---|
| LeNet-300-100 784-300-100 | Louizos *et al.* [2017] | 1.60% | 1.80% | - | 278-98-13 | 3.04 | 11% |
| | Louizos *et al.* [2018] | - | **1.40%** | 200 | 219-214-100 | 2.22 | 26% |
| | Louizos *et al.* [2018] | - | 1.80% | 200 | 266-88-33 | 3.06 | 10% |
| | Our method | 1.60% | 1.82% | **100** | 244-85-37 | **3.23** | **9%** |
| LeNet5 (MNIST) 20-50- 800-500 | Wen *et al.* [2016] | - | 1.00% | - | 3-12-800-500 | 1.04 | 25% |
| | Neklyudov *et al.* [2017] | - | 0.86% | - | 2-18-284-283 | 2.33 | 9% |
| | Louizos *et al.* [2017] | 0.90% | 1.00% | - | 5-10-76-16 | 12.8 | **7%** |
| | Louizos *et al.* [2018] | - | 0.90% | 200 | 20-25-45-462 | 2.48 | 50% |
| | Louizos *et al.* [2018] | - | 1.00% | 200 | 9-18-65-25 | **11.71** | 17% |
| | Our method | 0.78% | **0.80%** | **100** | 4-16-86-87 | 9.86 | 7% |
| VGG-like (CIFAR-10) 64x2-128x2- 256x3-512x7 | Li *et al.* [2017] | 6.75% | **6.60%** | **40** | 32-64-128-128-256-256-256-256-256-256-256-512 | 1.49 | 66% |
| | Neklyudov *et al.* [2017] | 7.20% | 7.50% | - | 64-62-128-126-234-155-31-79-73-9-59-73-56-27 | 4.03 | 43% |
| | Neklyudov *et al.* [2017] | 7.20% | 9.00% | - | 44-54-92-115-234-155-31-76-55-9-34-35-21-280 | 3.83 | 32% |
| | Our method | 7.60% | 8.50% | 150 | 37-41-91-89-156-140-74-81-54-51-44-46-48-52 | **4.72** | **23%** |

### 4.2 LeNet-300-100 and LeNet5 on MNIST Database

We first use MNIST dataset to evaluate the performance. Layer structure of LeNet-300-100 is [784, 300, 100, 10] and of LeNet5 is two [20,50] convolution layers, followed by two FC layers. The total number of trainable parameters of LeNet-300-100 and LeNet5 are 267K and 431K, respectively. Similar to previous works, we train reference models with standard training method with SGD optimizer, achieving accuracy of 1.72% and 0.78% respectively. In the pruning process, we use the softplus STE. The learning rate for $\mathcal{L}_1$ is scheduled from 1e-2 to 1e-3. During the training procedure, we observe that the final result is not sensitive to $\lambda$ and $\mu$ but the sparsification speed relies on $\mu$.

For neuron pruning, from Table 1, we can achieve the highest neuron compression rate(NCR) as 3.23 and the lowest FLOP usage percentage 9% comparing to original LeNet-300-100. For LeNet5, we are taking the lead in both the model accuracy 99.20% and the FLOP reduction rate 93%. For weight pruning, as we show in Table 4, our method applied to the LeNet-300-100 structure achieves the best compression rate of up to 80x while a 0.06% error increase. Note that all the other methods with compression rates greater than 60 have a minor accuracy drop while our method reaches the best accuracy. For LeNet5 model, we compare existing works with two reference models. For the first model with 0.78% error, we achieve 260x compression rate and 0.8% error. For the second model with 0.91% error, our method obtains a 310x compression rate with no accuracy drop.

Table 2: VGG-like CIFAR-10 Neuron Pruning

| Layer | Conv1 | Conv2 | Conv3 | Conv4 |
|---|---|---|---|---|
| Sparsity | 57.03% | 17.36% | 20.95% | 16.06% |
| FLOP | 57% | 37% | 45% | 49% |

| | Conv5 | Conv6 | Conv7 | Conv8 | Conv9 |
|---|---|---|---|---|---|
| | 10.76% | 4.67% | 5.30% | 1.52% | 0.39% |
| | 42% | 33% | 15% | 4.50% | 1.60% |

| | Conv10 | Conv11 | Conv12 | Conv13 |
|---|---|---|---|---|
| | 0.35% | 0.28% | 0.27% | 0.33% |
| | 1% | 0.85% | 0.77% | 0.84% |

Table 3: MobileNetV2(Top 1 Accuracy)

| FLOPs | Methods | FLOPs | Accuracy |
|---|---|---|---|
| 100M | Sandler et al. [2018] | 97M | 65.40% |
| | Yu et al. [2018] | 97M | 64.40% |
| | Yu and Huang [2019b] | 97M | 65.10% |
| | Our method | 102M | **66.83**% |
| 200M | Sandler et al. [2018] | 209M | 69.80% |
| | Tan et al. [2019] | 216M | 71.5% |
| | Yu and Huang [2019b] | 209M | 69.60% |
| | Wu et al. [2019] | 246M | 73% |
| | Yu and Huang [2019a] | 207M | 73% |
| | Our method | 209M | **73.32**% |
| 300M | Sandler et al. [2018] | 300M | 69.80% |
| | Tan et al. [2019] | 317M | 74% |
| | Yu and Huang [2019a] | 305M | **74.20%** |
| | Our method | 305M | 74.0% |

## 4.3 VGG-like on CIFAR-10

For VGG-like model, we use CIFAR-10 dataset to evaluate the performance. VGG-like is a standard convolution neural network with 13 convolutional layers followed by 2 FC layers (512 and 10 respectively). The total number of trainable parameters is 15M. Similar to previous works, we use the reference VGG-like model pre-trained with SGD with testing error 7.60%.

In this structure, we use L2-norm and L1-norm for $\mathcal{L}_1$ with hyperparameters 5e-5 and 1e-6, respectively. We evaluate both Leaky ReLU and Softplus STEs. Leaky ReLU gives a fast sparsification speed while Softplus shows a smooth convergence with approximately 1.5x running time. We suggest selecting the proper STE based on the time constraint.

For neuron pruning task, as shown in Table 1, our method reaches 23% FLOPs within 150 epochs. In Table 2, we show the layer-wise percentage FLOPs of VGG-16 structure. Our model achieves a higher sparsity at any layer compared to Li et al. [2017]. For weight pruning, our model reaches the highest 75x compression rate, with only moderate accuracy drop within 150 epochs of training.

## 4.4 AlexNet, ResNet-50 and MobileNet on ImageNet

Three models with ILSVRC12 dataset are also tested with our pruning method including 1M training images and 0.5M validation and testing images. AlexNet can be considered as deep since it contains 5 convolution layers and 3 FC layers. ResNet-50 consists of 16 convolution blocks with structure cfg=[3,4,6,3], plus one input and one output layer, and in total 25M parameters. For MobileNet, we

Table 4: Comparison of Different Weight Pruning Techniques

| Model | Methods | Error | CR |
|---|---|---|---|
| LeNet300-100 (MNIST) | Dong et al. [2017] | 1.76%→2.43% | 66.7 |
| | Ullrich et al. [2017] | 1.89%→1.94% | 64 |
| | Molchanov et al. [2017] | 1.64%→1.92% | 68 |
| | Our method | 1.72%→ **1.78%** | **80** |
| LeNet5 (MNIST) | Guo et al. [2016] | 0.91%→0.91% | 108 |
| | Ullrich et al. [2017] | 0.88%→0.97% | 162 |
| | Molchanov et al. [2017] | 0.80%→ **0.75%** | 280 |
| | Li et al. [2018] | 0.91%→0.91% | 298 |
| | Our method | 0.78%→0.80% | 260 |
| | Our method | 0.91%→0.91% | **310** |
| VGG-like (CIFAR-10) | Zhuang et al. [2018] | 6.01%→**5.43%** | 15.58 |
| | Zhu et al. [2018] | 6.42%→6.69% | 8.5 |
| | Molchanov et al. [2017] | 7.55%→7.55% | 65 |
| | Our method | 7.60%→7.82% | **75** |
| AlexNet (ILSVRC12) | Guo et al. [2016] | 43.42%→43.09% | 17.7 |
| | Srinivas et al. [2017] | 42.80%→**43.04%** | 10.3 |
| | Dong et al. [2017] | 43.30%→50.04% | 9.1 |
| | Our method | 43.26%→44.10% | **18.5** |
| ResNet50 (ILSVRC12) | Zhuang et al. [2018] | 23.99%→**25.05%** | 2.06 |
| | Our method | 25.10%→25.50% | **2.2** |

use its conventional MobileNet V2 (224×224) model with 310M FLOPs. The size of the dataset and also the complexity of the model clearly reveals the scalability of our method.

ResNet-50 is trained with a learning rate schedule from 1e-5 to 1e-6. Only L2 norm is applied, with $\lambda = 1e - 5$. Note that the identity connections alleviate the need to add layer-wise learning rate since the gradient to the first several layers is enough to pull the auxiliary parameters. The learning rate for AlexNet is 1e-3 and for MobileNet V2 is 1e-5. We split the training data into 1:1 for weight update and auxiliary parameter update respectively. Once the desired FLOPs is reached, we use all training data to fine tune the model.

For neuron pruning, we evaluate our method on compact MobileNet V2 with less redundancy, and compare with the state-of-art methods in different FLOPs levels, in Table 3. Our method achieves similar error at 300M level and outperforms others at extreme level(200M and 100M). For ResNet at 600M FLOPs, the top-1 error is 27.6%. For weight pruning, the results in Table 4 show that our method on AlexNet model achieves 18.5x compression rate and 0.84% accuracy drop. For ResNet-50, we get 2.2x compression rate with only 0.4% accuracy drop.

## 4.5 Ablation Study

We show the sparsity and accuracy are not sensitive to hyperparameters, taking weight pruning with VGG-like on CIFAR-10 as an example. In Fig. 2(a), we set the learning rate of auxiliary parameters to 1e-2, 1e-1 and 5e-1. From the result we observe that all three settings converge to similar compression ratio with different sparsification speed. In Fig. 2(b), the accuracy with higher learning rate drops faster, but the final gap is less than 0.1%. In Fig. 2(c), we show the compression ratio versus accuracy plot with proposed update in Eq. 7 and regular BNN update. The regular BNN update becomes non-stable after 30x CR, and accuracy drops sharply afterward. With the proposed update rule, accuracy is more stable and with lower variance until 80x. We've also included the comparison on choosing different STE functions and learning rates for VGG like model on CIFAR10 in Fig. 2(d). Softplus STE achieves the best result while converges slower than LeakyReLU STE, which achieves slightly lower CR. The linear STE however, yields worst CR and slower convergence speed.

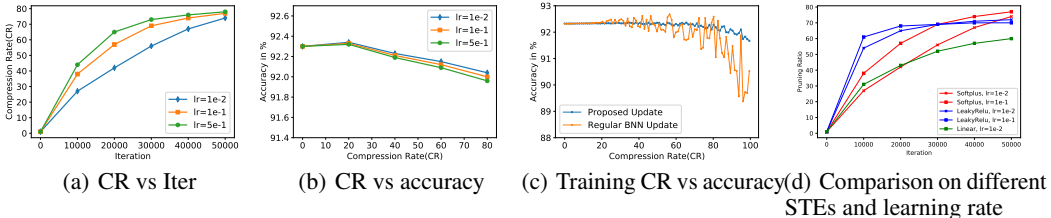

(a) CR vs Iter     (b) CR vs accuracy     (c) Training CR vs accuracy (d) Comparison on different STEs and learning rate

Figure 2: Illustration of Hyperparameter Sensitivity

## 4.6 Training From Scratch

Apart from sparsification on pre-trained models, our method can support training sparse network from scratch. We evaluate our method through training LeNet5 from scratch. All the weights are randomly initialized as usual while the auxiliary parameters are initialized as $m_{ij} \sim Gaussian(0.1, 0.05)$. The initial learning rate is set to 1e-3 and gradually decreased to 1e-5. The final model we obtain has an error of 0.95% with a 168x compression rate.

# 5 Conclusions

In this paper, we propose to automatically prune deep neural networks by regularizing auxiliary parameters instead of original weights values. The auxiliary parameters are not sensitive to hyperparameters and are more robust to noise during training. We also design a gradient-based update rule for auxiliary parameters and analyze the benefits. In addition, we combine sparse regularizers and weight regularization to accelerate the sparsification process. Extensive experiments show that our method achieves the state-of-the-art sparsity in both weight pruning and neuron pruning compared with existing approaches. Moreover, our model also supports training from scratch and can reach a comparable sparsity.

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
