[Reviews · NeurIPS 2019]

Reviewer 1



After Rebuttal: The new test results are very welcome and indeed show that the proposed method is on-par with the most recent SOTA works in terms or compression rates, which greatly enhances its significance in my opinion. As a result I increased my score. While using masks is not entirely novel, having auxiliary parameters to decouple the pruning and learning processes is I think quite important and seems to be novel. While the lack of theoretical contributions makes the suitability of the paper to NeurIPS debatable, I think this still is a good paper and I wouldn't mind it being accepted. **************************** Literature review: Related work covered is not very extensive - it mentions only pruning related methods and not neural architecture search, quantization, low rank approximation or other kinds of attempts at efficient deep learning. Since the paper focuses on pruning only this might be fine. I'd suggest adding Targeted Dropout by Gomez et al. here. There are also some recent important papers in this area, such as AutoSlim by Yu et al. and the newer version of the Targeted Dropout paper "Learning Sparse Networks Using Targeted Dropout". What the paper is lacking is a proper mention of structured vs. unstructured pruning (neuron vs. weight pruning in the paper), and the implications of the ground breaking paper "Rethinking the value of pruning" by Liu et al, especially for structured pruning. Mentioning these papers and positioning the paper accordingly would result in a better paper. Methodology: I thought the explanation of the method was excellent. The approach is well justified, and is explained very clearly, leaving no room for confusion. The separation of gradient rules for the auxiliary and network parameters is a very good idea. The paper experiments with different forms of STE and shows that the default linear option is not good, and suggests using leaky ReLU or softplus STE, which is another nice contribution in my opinion. The careful analysis of the gradients of weights and auxiliary parameters is also valuable. The update rule forces the gradients of w_ij and m_ij to take the same signs, dubbed "correlation consistency". The proofs are provided and are easy to follow (I think Eq 10 misses an 'sgn') Recoverability of the pruning is another good property. Unlike using sparse regularizers that kill the weights completely, which is hard to recover from, this network retains the weights and only turns them on or off with the auxiliary parameters, which makes it possible to recover them. This is similar to a recent paper called "Discovering Neural Wirings" by Wortsman et al, where entire nodes are turned on or off this way. Experiments: Even though I like the method and the paper is quite excellent I think, the experiments are a bit lacking. One big problem is the reliance on the LeNet based models on MNIST. Compression rates of LeNet for MNIST are almost meaningless, because even much smaller networks can already reach much higher accuracies. For instance, a network with 16 and 32 filters in 2 conv layers, followed by only a single hidden layer of 64 neurons combined with dropout can already reach 99.5%. This network has 114K parameters. LeNet5-20-50-800-500 is huge in comparison with 2.4M parameters (the paper says 431K which is I think wrong) and has a larger base error. Even a compression rate of 10 is therefore meaningless. This effectively makes half of the supplied experiments useless. I'd suggest that the authors focus on more modern CNNs as the recent papers do. I think the papers compared against are rather old. There are many SOTA papers from the last two years worth comparing against. One good example is AutoSlim paper, which compares itself against many other recent works, so that would be a good starting point. Currently the condition the experiments section is in is very underwhelming and unconvincing, which is unfortunate for an otherwise very good paper. I also think that the paper needs more extensive ablation studies to show the effect of correlation consistency for auxiliary updates. What happens if it is not forced? What changes if different types of STE's are used? In general I've found the paper quite interesting and thought it deserves acceptance. However lack of proper experiments with newer works and the amount of energy and text spent on LeNets made me reduce my score.

Reviewer 2



Originality: the paper proposed pruning deep learning weights based on auxiliary parameters, which seems to be novel. Quality: the work is generally complete with sufficient motivation, discussion of the benefit of the approach and numerical studies. The work lacks the theoretical justification of the approach, which is a weakness. Clarity: the paper is generally clear, with a few minor typos. For example, R1 was first used in (9) and then later defined in (12), which could lead to confusion in readers. Significance: I believe this proposal could be of practical use for practitioners. The work will be more complete if there are theory that justifies the proposal. ============================================================= Comments read.

Reviewer 3



Originality: The novelty lies in two aspects: 1) Reformulation of the pruning problem as a mask learning problem 2) Modify the corresponding gradient update rule. The novelty is not good enough for a NeurIPS paper in the following sense. a) The masking formulation has been adopted by [1] to learn sparse NN. Actually, this formulation has long been adopted by the LASSO variants in traditional structural learning say [2] and the references therein. b) The proposed update rule is a straightforward binarization of the weights. Moreover how to calculate the partial derivation $\frac{\partial{h(m_{ij})}}{\partial{m_{ij}}}$ Reference [1] Louizos C, Welling M, Kingma D P. Learning Sparse Neural Networks through $ L_0 $ Regularization[J]. arXiv preprint arXiv:1712.01312, 2017. [2] Frecon J, Salzo S, Pontil M. Bilevel learning of the group lasso structure[C]//Advances in Neural Information Processing Systems. 2018: 8301-8311. Clarity: The motivation and research goal are clearly written, and the organization follows good logic. Quality: Presentation needs to be improved, both mathematically and literally. Some typical issues are listed below: 1) 4) and 5) in the last paragraph of the introduction section should not be understood as major contributions of this paper. They only validate the significance of the proposed method. The authors might want to merge them into 3). 2) It is misleading to call the proposed update direction a gradient, especially in the theoretical discussions in Sec. 3.3.. Also, the notation $\partial{\mathcal{L}_2}{m_{ij}}$ should be changed. 3) In Sec.3.3, the name ‘sensitivity consistency’ does not make sense to me. Replacing w_ij with sign(w_ij) makes m_ij insensitive toward the weight magnitude, rather than correcting the sensitivity. 4) Eq.(3) and Eq.(4)-(5) are inconsistent, the authors might want to add $\lambda\mathcal{R}(W)$ on Eq.(3). 5) The second paragraph of Sec. 3.4 why weight decay damages the recoverability is not clearly explained. 6) In page 3, Line 1: The sentence 'The goal is to find the minimum subset w \in \mathcal{W} that reduces the model accuracy.' should be rephrased. The Goal is to preserve the model accuracy under the pruning process. Significance: The experimental results shown significant improvement on a variety of datasets and network architectures. This paper presents an interesting work on network pruning. However, balancing pros and cons, I think this paper is not good enough to be accepted as a NeurIPS paper, especially in terms of the originality and quality issues.

[Author Response · NeurIPS 2019]

**Review1:** Thank you for the review and we appreciate the insightful suggestions. We will improve the paper accordingly:

In the related work section, we will first introduce structured and unstructured pruning and summarize existing papers
in these two categories, respectively. We also will include two latest papers as follows:

Gomez *et al.* [2019] proposed to keep neurons with high magnitude and prune only neurons with smaller magnitude in
a stochastic way. The accuracy is maintained by reducing the dependency of important neurons on unimportant neurons.
Liu *et al.* [2018] did comprehensive experiments showing that training-from-scratch on the right sparse architecture
yields better results than pruning from pre-trained models. Such a result implies that the pruning process is actually
finding the right network structure, thus bridging the gap between pruning and architecture search.

Our paper can be further positioned as a structure learning procedure where a neuron's importance is learned by the
attached trainable auxiliary parameter. Unimportant neurons will be pruned if the corresponding auxiliary parameter
decreases below a threshold, and important neurons will be recovered otherwise.

For the experiment part, thank you for referring the AutoSlim paper. We did additional experimental comparison on
MobileNet v1 and v2 with the AutoSlim paper. Our model achieves better performance than AutoSlim under extreme
cases, where FLOPs are compressed down to 150M and 209M, respectively. We will include more neuron pruning
results and discussion on ResNet50, SuffleNet, and MNasNet to replace the LeNet5 part in the camera ready version.
We also include the performance of different STEs on VGG-like below, and will add more ablation discussion and
results. Compared to Softplus STE, LeakyRelu converge faster but with lower CR, and linear STE converge slower
with lower CR. In addition, figure 2(c) in the paper shows the comparison with and without forcing the sign.

| Model | MobileNet v1 | | | MobileNet v2 | | |
|---|---|---|---|---|---|---|
| | Base | AutoSlim | Ours | Base | AutoSlim | Ours |
| FLOPs | 325M | 325M | 325M | 300M | 305M | 300M |
| Top1-Err | 31.6 | **28.5** | 28.8 | 28.2 | **25.8** | 26.1 |
| FLOPs | 150M | 150M | 150M | 209M | 207M | 209M |
| Top1-Err | 36.7 | 32.1 | **31.8** | 30.2 | 27 | **26.7** |

**Review2:** Thank you for the comments of the paper especially on the clarity and potential improvement. We will
mention before Eq. (9) that $R_n$ refers to n-norm regularization. We included some theoretical analysis on the correlation
gradient direction consistency in the paper which ensures the effectiveness of gradient descent. We will also include the
necessary convergence discussion in the appendix.

**Review3:** We would like to thank reviewer for the valuable thoughts and detailed comments on the paper presentation.

Originality: The significant difference from other mask based pruning lies on the weak-correlation between the weights
and auxiliary parameters. We decoupled the correlation between update rule and weight magnitude which is harmful to
pruning task, as is discussed in Sec3.3 and figure 2(c). Theoretical analysis and experimental results show such weak
correlation leads to a more compact and more accurate model.

a) We agree that the masking formulation has been adopted by existing papers to learn sparse NNs, but optimizing the
binary masks effectively, stably and efficiently requires significant research efforts. The modified update rule results in
a better local minima of the auxiliary parameter network for the pruning task. Moreover, we have compared with [1] in
the introduction section that, "In contrast to Louizos et al.[2018], our method avoids inefficient and high variance single
step Monte-Carlo sampling and places no assumptions on the prior distribution". We also show in Table 1 that our model
significantly outperforms [1]. b) The update rule is carefully designed as mentioned above. The way of calculating the
partial derivations follow coarse gradients of predefined straight through estimators as discussed in Sec3.2. Specifically,
our Tensorflow implementation employs gradient overriding function for coarse gradient calculation.

Quality: 1) We agree that the high scalability and training-from-scratch ability mentioned in 4) and 5) validate the
significance of the proposed method. We will reorganize the list of contributions based on your nice suggestion. 2) The
update rule is not following an accurate gradient w.r.t. $m$, but a direction that forms an acute angle with the original
gradient direction. Here we kind of abuse the word "gradient" since it is the actual direction that auxiliary parameters
will be updated on. We will rephrase it as "updating rule" or "modified gradient" to avoid potential misunderstanding.
3) Thank you for pointing out the inaccurate naming of "sensitivity consistency". We will rename it "sensitivity
decoupling", since the update rule of auxiliary parameters is decoupled with the magnitude of weights. 4) Thank you
for your suggestion to improve the equations' consistency. We will add the regularization term $\lambda \mathcal{R}(\tilde{W})$ of weight $\tilde{W}$ to
Eq.(3). 5) We will add the following additional explanation on recoverable pruning to Sec3.4: Recovering a weight $w_{ij}$
requires the gradient of $m_{ij}$ moving toward positive direction. The weight decay will decrease the magnitude of $w_{ij}$
and provide a negative gradient to $m_{ij}$, which reduces the recoverability. 6) There is a typo in the sentence. We will
rephrase it to "The goal is to find the minimum subset $\{w : w \in \mathcal{W}\}$ that preserves the model accuracy."

[Meta-Review · NeurIPS 2019]

Reviewers discussed on novelty and concluded that, although the main idea is not very new, a few small novel ideas are important and useful. Also they all agreed that experiments showed practical significance.